# SELECTNOISE: Unsupervised Noise Injection to Enable Zero-Shot Machine Translation for Extremely Low-resource Languages

**Maharaj Brahma**[*]     **Kaushal Kumar Maurya**[*]     **Maunendra Sankar Desarkar**

Natural Language and Information Processing Lab (NLIP)

Indian Institute of Technology Hyderabad

Hyderabad, India

{cs23resch01004,cs18resch11003}@iith.ac.in, maunendra@cse.iith.ac.in

## Abstract

In this work, we focus on the task of machine translation (MT) from extremely low-resource language (ELRLs) to English. The unavailability of parallel data, lack of representation from large multilingual pre-trained models, and limited monolingual data hinder the development of MT systems for ELRLs. However, many ELRLs often share lexical similarities with high-resource languages (HRLs) due to factors such as dialectical variations, geographical proximity, and language structure. We utilize this property to improve cross-lingual signals from closely related HRL to enable MT for ELRLs. Specifically, we propose a novel unsupervised approach, SELECTNOISE, based on *selective candidate extraction* and *noise injection* to generate noisy HRLs training data. The noise injection acts as a regularizer, and the model trained with noisy data learns to handle lexical variations such as spelling, grammar, and vocabulary changes, leading to improved cross-lingual transfer to ELRLs. The selective candidates are extracted using BPE merge operations and edit operations, and noise injection is performed using greedy, top-p, and top-k sampling strategies. We evaluate the proposed model on 12 ELRLs from the FLORES-200 benchmark in a zero-shot setting across two language families. The proposed model outperformed all the strong baselines, demonstrating its efficacy. It has comparable performance with the supervised noise injection model. Our code and model are publicly available[1].

## 1 Introduction

The modern neural machine translation (NMT; Aharoni et al. (2019); Garcia et al. (2021); Siddhant et al. (2022)) has achieved remarkable performance for many languages, but their performance heavily relies on the availability of large parallel or mono-

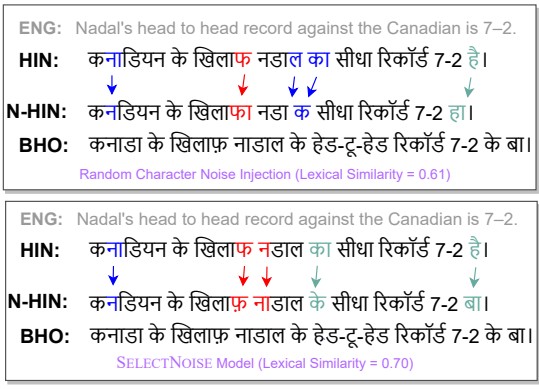

Figure 1: Illustration of character noise injection with random baseline (Aepli and Sennrich, 2022) and propose SELECTNOISE model. The SELECTNOISE enhances lexical similarity between noisy HRL (N-HIN) and ELRL (BHO). ENG: *English*, HIN: *Hindi*, N-HIN: *Noisy Hindi* and BHO: *Bhojpuri* languages. Red: *Insertion*, Blue: *Deletion* and Green: *Substitution* operations.

lingual corpora (Koehn and Knowles, 2017). However, the linguistic diversity across the world is vast, with over 7000 languages spoken[2]. This linguistic landscape includes a long tail of languages (Joshi et al., 2020) that face significant challenges due to the lack of available resources for model development and are referred as extremely low-resource languages (ELRLs). ELRLs present unique challenges for the development of MT systems as they lack parallel datasets, are excluded from large multilingual pre-trained language models, and possess limited monolingual datasets. Consequently, the majority of research efforts in NMT have primarily focused on resource-rich languages (Bender, 2019), leaving ELRLs with limited attention and fewer viable solutions. Towards these concerns, this work is positioned as a step towards enabling MT technology for ELRLs. Primarily focused on *zero-shot* setting for scalability.

More recently, there has been active research to develop MT systems for LRLs. One direction is

---

[*]Equal contributions

[1]code and model checkpoints link: https://github.com/maharajbrahma/selectnoise

[2]https://www.ethnologue.com/

multilingual training-based models (Aharoni et al., 2019; Garcia et al., 2021; Siddhant et al., 2022). These models are trained with multiple HRL languages, enabling cross-lingual transfer capabilities to improve translation performance for LRLs. Another line of work focuses on data augmentation techniques (Sennrich et al., 2016a; Wang et al., 2018) to generate more training data. However, these methods do not fully exploit the lexical similarity between HRLs and ELRLs. Many HRLs and ELRLs exhibit surface-level lexical similarities due to dialect variations, loan words, and geographical proximity (Khemchandani et al., 2021). For example, the word "Monday" is *somvar* in Hindi and *somar* in Bhojpuri. They are lexically very similar. To leverage this lexical similarity, recent studies have explored techniques like learning overlapping vocabulary (Patil et al., 2022) or injecting random noise (Aepli and Sennrich, 2022; Blaschke et al., 2023) in HRLs to resemble LRLs. These methods are only evaluated for natural language understanding tasks (NLU) and not for MT, which is a more challenging task. Inspired by these advancements, in this paper, we propose a novel unsupervised noise injection approach to develop an MT system for ELRLs.

The proposed model is based on character noise injection and consists of two stages: *Selective Candidate Extraction* and *Noise Injection*. In the selective candidate extraction phase, candidate characters are extracted in an unsupervised manner using small monolingual data from closely related HRL and ELRLs. It relies on BPE merge operations and edit operations that take into account lexical similarity and linguistic properties. In the noise injection phase, noise is injected into the source side of parallel data of HRL using greedy, top-k, and top-p sampling algorithms. This noise injection serves as a regularizer and a model trained with this noisy HRL data enhances robustness to spelling, grammar, or vocabulary changes and facilitates improved cross-lingual transfer for ELRLs. The evaluation is done in the *zero-shot* setting, ensuring scalability. The proposed model is referred as the SELECTNOISE: *Unsupervised Selective Noise Injection* model. Fig. 1 illustrates the effect of noise injection with SELECTNOISE model. In this paper, we investigate two hypotheses: (a) *the selective noise injection model is expected to outperform random noise injection*, and (b) *the performance of the selective (unsupervised) noise injection model*

*should be comparable to the supervised noise injection model.*

Our key contributions are: (1) We propose a novel unsupervised selective character noise injection approach, SELECTNOISE, to enable and improve MT for ELRLs to English. The injection of selective candidate noise facilitates better cross-lingual transfer for ELRLs in the zero-shot setting. (2) We have developed an unsupervised mechanism to *extract candidate characters* based on BPE merge operations and edit operations. Furthermore, the *noise injection* employs greedy, top-k, and top-p sampling techniques, ensuring diversity. (3) We evaluated the model's performance using 12 ELRLs from the FLORES-200 evaluation set across two typologically diverse language families. Evaluations were conducted with both automated and human evaluation metrics. (4) The proposed model outperformed all baselines and has comparable performance with the supervised selective noise injection-based MT model. Additionally, we performed several analyses to demonstrate the robustness of the proposed model.

## 2 Methodology

This section presents the details of the proposed SELECTNOISE model. As discussed in section 1, the SELECTNOISE model has two stages: *Selective Candidate Extraction* and *Noise Injection*. In the selective candidate extraction stage, the noise injection candidate characters are extracted through an unsupervised approach. Specifically, we consider small monolingual data for HRL ($\mathcal{D}_{\mathcal{H}}$) and for related (lexically similar) LRLs ($\mathcal{D}_{\mathcal{L}}$). $\mathcal{D}_{\mathcal{L}}$ comprises small monolingual datasets from multiple extremely ELRLs. During the process of building the Byte Pair Encoding (BPE) vocabulary, we extract the BPE operations separately for the HRL ($\mathcal{B}_{\mathcal{H}}$) and ELRLs ($\mathcal{B}_{\mathcal{L}}$). Next, we design an algorithm $\mathcal{A}$ to extract selective candidate characters from $\mathcal{B}_{\mathcal{H}}$ and $\mathcal{B}_{\mathcal{L}}$ and store them in candidate pool $\mathcal{S}_{\mathcal{C}}$. Candidates are extracted with an approach inspired by edit-operations. In other words, we obtain $\mathcal{S}_{\mathcal{C}}$ as a result of $\mathcal{A}$ ($\mathcal{B}_{\mathcal{H}}$, $\mathcal{B}_{\mathcal{L}}$). In noise injection stage, the candidates are sampled from $\mathcal{S}_{\mathcal{C}}$ and injected into the source sentences of HRL corpus ($\mathcal{H}$) from the large parallel data $\mathcal{P}_{\mathcal{H}} = \{(h,e)|lang(h) = \mathcal{H}, lang(e) = En\}$. The injections are done using a noise function $\eta$, resulting in a noise-injected (augmented) parallel data: $\hat{\mathcal{P}}_{\mathcal{H}} = \{(\hat{h},e)|lang(\hat{h}) = \hat{\mathcal{H}}, lang(e) =$

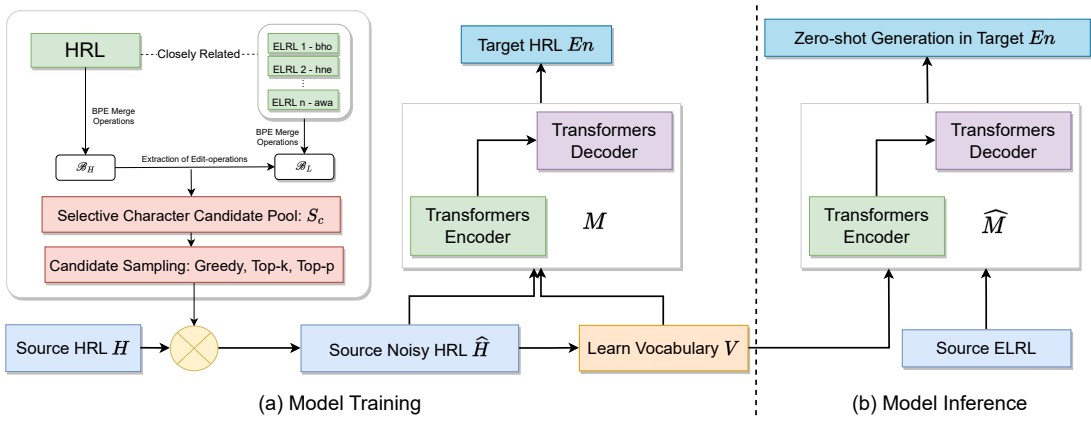

Figure 2: Overview of the proposed SELECTNOISE model for extremely low-resource MT

$En\}$, where $\hat{\mathcal{H}} = \eta(\mathcal{H})$. The $\hat{\mathcal{H}}$ acts as proxy training data for ELRLs. A BPE vocabulary $\mathcal{V}$ is learned with $\hat{\mathcal{H}}$. Then, we train the standard encoder-decoder transformers model ($\mathcal{M}$; Vaswani et al. (2017b) from scratch with $\hat{\mathcal{H}}$ and $\mathcal{V}$ to obtain trained model $\hat{\mathcal{M}}$. Finally, zero-shot evaluation is done for ELRLs with $\hat{\mathcal{M}}$ and $\mathcal{V}$. Now, we present details of each model component. The overview of the proposed SELECTNOISE model is depicted in Figure 2.

### 2.1 SELECTNOISE: Unsupervised Noise Injection

The formal procedure for unsupervised noise injection is presented in Algorithm 1. In the next subsections, we will deep dive into each stage of the proposed model in details:

#### 2.1.1 Selective Candidate Extraction

The first stage in the proposed approach involves extracting candidate characters that will subsequently be utilized for noise injection. Given $\mathcal{D}_{\mathcal{H}}$ and $\mathcal{D}_{\mathcal{L}}$, we extract all BPE merge operations $\mathcal{B}_{\mathcal{H}}$ and $\mathcal{B}_{\mathcal{L}}$, respectively. Each merge operation consists of tuples $\langle (p, q) \rangle \in \mathcal{B}_{\mathcal{H}}$ and $\langle (r, s) \rangle \in \mathcal{B}_{\mathcal{H}}$. We pair each merge tuple of $\mathcal{B}_{\mathcal{H}}$ with each tuple of $\mathcal{B}_{\mathcal{L}}$ (i.e., cartesian setup). If $\mathcal{B}_{\mathcal{H}}$ and $\mathcal{B}_{\mathcal{L}}$ have $n$ and $m$ merge operations, respectively, we obtain a total of $t = m \cdot n$ pairs. We consider only those pairs where either $p$ and $r$ or $q$ and $s$ are the same while discarding the rest. For the considered tuples $\langle (p, q), (r, s) \rangle$, we calculate the character-level edit-distance operations between non-similar elements of the tuple. For instance, if $p$ and $r$ are the same, the edit operations are obtained using $q$ and $s$ elements. These operations are collected in the candidate pool $\mathcal{S}_c$, which includes *insertions*, *deletions*, and *substitutions*, and are referred to as the *selective candidates*.

As discussed, the extracted selective candidates are stored in the candidate pool $\mathcal{S}c$, a dictionary data structure encompassing HRL and ELRL characters. The $\mathcal{S}c$ consists of HRL characters, ELRL characters, edit operations, and their respective frequencies. An element of $\mathcal{S}c$ has following template: $c_i : \{I : f_{ins}, D : f_{del}, S : \{c'_1 : f_1, c'_2 : f_2, c'_k : f_k\}\}$. The operations are: *insertion* ($I$), *deletion* ($D$) and *substitution* ($S$). The character $c_i$ represents the $i^{th}$ element of $\mathcal{S}_c$, which is an HRL character, $c'_1 \ldots c'_k$ denote the corresponding substituting candidates from ELRLs and $f$ is the associated frequencies. A few examples of selective candidate extraction are illustrated in Fig. 3. Sample candidate pool ($\mathcal{S}_c$) is shown in Fig. 6.

Intuitively, with this candidate pool $\mathcal{S}_c$, we have learned transformative entities that can be used to resemble an HRL to lexically similar ELRLs, which results in bridging the lexical gap between HRL and LRLs. Training with such modified HRL data enhances the effectiveness of cross-lingual transfer signals for ELRLs. As candidates are extracted by considering the vocabulary word-formation strategy from BPE and edit operations, they indirectly consider the linguistic cues/information.

#### 2.1.2 Noise Injection to HRL

In the second stage, we sample selective candidates from $\mathcal{S}_c$ and inject into the source sentences of HRL corpus ($\mathcal{H}$) from the parallel dataset $\mathcal{P}_{\mathcal{H}} = \{(h, e)|lang(h) = \mathcal{H}, lang(e) = En\}$ using a noise function $\eta$, resulting in a noise injected (augmented) parallel dataset $\hat{\mathcal{P}_{\mathcal{H}}} = \{(\hat{h}, e)|lang(\hat{h}) = \hat{\mathcal{H}}, lang(e) = En\}$, where $\hat{\mathcal{H}} = \eta(\mathcal{H})$. Details of

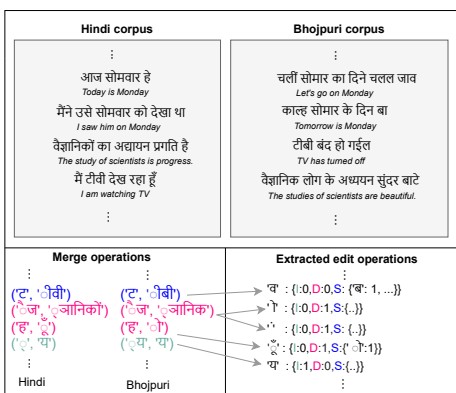

Figure 3: Illustration of selective candidates extraction for noise injection that utilizes BPE merge and edit operations. Here *I*, *D*, and *S* indicate insertion, deletion, and substitution respectively. Frequencies are associated with operations. 0 indicates the corresponding edit operation was not extracted.

the noise function and candidate sampling strategy are presented below:

**Noise Function:** The noise injection function ($\eta$) is designed as follows: Initially, we randomly select 5%-10%[3] of character indices from a sentence $s \in \mathcal{H}$. Subsequently, we uniformly choose between *insertion*, *deletion*, or *substitution* operations with equal probability. If the selected operation is insertion or substitution, we sample a candidate character from $\mathcal{S}_c$ to perform the noise injection operation. For deletion, the charter is simply deleted. These steps are repeated for all sentences in $\mathcal{H}$ to obtain the final $\hat{\mathcal{H}}$.

**Candidate Character Sampling:** While noise injection for deletion operation, we simply delete the character. For insertion and substitution, we sample the candidate character for injection from $\mathcal{S}_c$ using the *greedy*, *top-p* (nucleus), and *top-k* sampling techniques inspired by decoding algorithms commonly employed in NLG (Holtzman et al., 2019). Before applying these sampling techniques, the frequencies of the candidate characters are transformed into probability scores using the softmax operation. Intuitively, with the sampling technique, we aim to explore not only frequent candidate characters but also diverse candidates.

The performance of any learning model depends on the quality of the training data. The presence of noise hampers the learning, and the outputs of the learned model exhibit the different nuances of the noise present in the data. In our specific case: (i) We train a model using data that contains noise, resulting in the model's increased robustness to minor lexical variations in different languages, par-

---

[3]after conducting several ablation experiments, this range provides the best performance

ticularly those related to ELRLs. (ii) The noise is added for a small portion of characters (5-10%), making the HRLs training data closely resemble how sentences appear in ELRLs. As a result, the model is able to do a robust cross-lingual transfer to the ELRL in a zero-shot setting. In another perspective, the injection of noise acts as a regularizer (Aepli and Sennrich, 2022), contributing to an overall enhancement in the model's performance.

## 2.2 Supervised Noise Injection

We have also investigated in a supervised setting akin to the proposed SELECTNOISE approach. The key distinction lies in how the candidate pool $\mathcal{S}sc$ is derived from a limited parallel dataset between HRL and ELRLs. For each HRL and ELRL pair,

---

**Algorithm 1** SELECTNOISE: Unsupervised Noise Injection

---

**Require: [Inputs]** HRL monolingual data $\mathcal{D}_{\mathcal{H}}$; closely related ELRLs monolingual data $\mathcal{D}_{\mathcal{L}}$; number of merge operations $\mathcal{M}_{\mathcal{O}}$; HRL parallel data $\mathcal{P}_{\mathcal{H}}(\mathcal{H}, En)$; Noise injection percentage range [$p_1\%$ - $p_2\%$]; candidate sampling strategy $\mathcal{S}_{\mathcal{M}}$; EXTRACTSELECTIVECANDS ($\mathcal{A}$)
**Ensure: [Output]** Noisy source HRL $\hat{\mathcal{H}}$
  $\mathcal{S}_c$ = EXTRACTSELECTIVECANDS($\mathcal{D}_{\mathcal{H}}, \mathcal{D}_{\mathcal{L}}, \mathcal{M}_{\mathcal{O}}$)
  **for** sentence $s$ in $\mathcal{H}$ **do**
    $idxs \leftarrow$ randomly select [$p_1\%$ - $p_2\%$] indices of $s$
    **for** $idx$ in $idxs$ **do**
      $ops \leftarrow$ randomly sample operation {*insert*, *delete*, *substitute*}
      **if** $ops$ equals *delete* **then**
        Remove character at index $idx$
      **end if**
      **if** $ops$ equals *Insert* **or** $ops$ equals *substitute* **then**
        $c$ = sample candidate char, i.e., $\mathcal{S}_{\mathcal{M}}(\mathcal{S}_c, ops)$
        Perform operation $ops$ at index $idx$ with $c$
      **end if**
    **end for**
  **end for**
  **procedure** EXTRACTSELECTIVECANDS($\mathcal{D}_{\mathcal{H}}, \mathcal{D}_{\mathcal{L}}, \mathcal{M}_{\mathcal{O}}$)
    Initialize candidate pool $\mathcal{S}_c \leftarrow \emptyset$ to store candidates
    Compute merge operations $\mathcal{B}_{\mathcal{H}}$ = BPE($\mathcal{D}_{\mathcal{H}}, \mathcal{M}_{\mathcal{O}}$)
    Compute merge operations $\mathcal{B}_{\mathcal{L}}$ = BPE($\mathcal{D}_{\mathcal{L}}, \mathcal{M}_{\mathcal{O}}$)
    **for** $n$ in $\mathcal{B}_{\mathcal{H}}$ **do**
      **for** $m$ in $\mathcal{B}_{\mathcal{L}}$ **do**
        ▷ where $n$ = tuple $\langle (p, q) \rangle$, $m$ = tuple $\langle (r, s) \rangle$
        **if** $n$ equals $m$ **or** ($p$ not equals $r$ **and** $q$ not equals $s$) **then**
          No operation is performed with $n$ and $m$
        **end if**
        **if** $p$ equals $r$ **then**
          Compute edit-operations($q$, $s$) & update $\mathcal{S}_c$
        **end if**
        **if** $q$ equals $s$ **then**
          Compute edit-operations($p$, $r$) & update $\mathcal{S}_c$
        **end if**
      **end for**
    **end for**
    Return $\mathcal{S}_c$
  **end procedure**

**Algorithm 2** Supervised Noise Injection

---

**Require: [Inputs]** joint parallel data for all considered HRL-ELRL pairs $\mathcal{P}_S(\mathcal{S}, \mathcal{E}_L)$; HRL parallel data $\mathcal{P}_\mathcal{H}(\mathcal{H}, En)$; noise injection percentage range [$p_1\%$ - $p_2\%$]; candidate sampling strategy $\mathcal{S}_\mathcal{M}$

**Ensure: [Output]** Noisy source HRL $\hat{\mathcal{H}}$

  $\mathcal{S}_{sc}$ = SUPEXTRACTSELECTIVECANDS($\mathcal{P}_S$)

  **for** sentence $s$ in $\mathcal{H}$ **do**

    $idxs \leftarrow$ randomly select [$p_1\%$ - $p_2\%$] indices of $s$

    **for** $idx$ in $idxs$ **do**

      $ops \leftarrow$ randomly sample operation {*insert, delete, substitute*}

      **if** $ops$ equals *delete* **then**

        Remove character at index $idx$

      **end if**

      **if** $ops$ equals *Insert* **or** $ops$ equals *substitute* **then**

        $c$ = sample candidate char, i.e., $\mathcal{S}_\mathcal{M}(\mathcal{S}_{sc}, ops)$

        Perform operation $ops$ at index $idx$ with $c$

      **end if**

    **end for**

  **end for**

  **procedure** SUPEXTRACTSELECTIVECANDS($\mathcal{P}_S$)

    Initialize candidate pool $\mathcal{S}_{sc} \leftarrow \emptyset$ to store candidates

    **for** each $\langle (s,e) \rangle$ in $\mathcal{P}_S$ **do**

      Compute edit-operations($s, e$) & update $\mathcal{S}_{sc}$

    **end for**

    return $\mathcal{S}_{sc}$

  **end procedure**

---

we extract a candidate set using edit operations and subsequently combine all the candidate sets in $\mathcal{S}c$. The rest of the modeling steps are similar to the SELECTNOISE. We hypothesize that the unsupervised method should exhibit competitive performance compared to the supervised approach. In the supervised candidate extraction, we assume the availability of a limited amount of parallel data of approximately 1000 examples. A formal algorithm outlining in the Algorithm 2.

## 2.3 Model Training and Zero-shot Evaluation

The stranded encoder-decoder transformers model ($\mathcal{M}$) is trained from scratch using the noisy high-resource parallel dataset $\hat{\mathcal{P}}_\mathcal{H}$ and $\mathcal{V}$ to obtain a trained model $\hat{\mathcal{M}}$. Where $\mathcal{V}$ is learned BPE vocabulary with $\hat{\mathcal{P}}_\mathcal{H}$. Subsequently, we use $\hat{\mathcal{M}}$ to perform zero-shot generation for ELRLs. We have not used any parallel training data for ELRLs and directly employ $\hat{\mathcal{M}}$ for inference, making this modeling setup zero-shot. The trained model transfers knowledge across languages, enabling coherent and meaningful translation for ELRLs.

## 3 Experimental Setup

We designed our experimental setup to address the following set of questions: (1) Does noise injection improve performance for NLG tasks, i.e., MT in

our case? (2) Does selective noise injection with the proposed SELECTNOISE model outperform the random noise injection model (Aepli and Sennrich, 2022)? (3) Does the model's performance persist across different language families? and (4) Does the unsupervised SELECTNOISE model demonstrate competitive performance with supervised approach? Based on these research questions, we have designed our experimental setup.

## 3.1 Datasets

The primary constraint of the proposed approach is to select closely related HRLs and ELRLs. With this criterion in mind, we have chosen two language families: *Indo-Aryan* and *Romance*. Within the Indo-Aryan family, we have selected Hindi (hi) as the HRL and 8 ELRLs were Awadhi (awa), Bhojpuri (bho), Chhattisgarhi (hne), Kashmiri (kas), Magahi (mag), Maithili (mai), Nepali (npi), and Sanskrit (san), based on their lexical similarity. For the Romance family, Spanish (es) served as the HRL, and the 4 ELRLs were Asturian (ast), Catalan (cat), Galician (glg), and Occitan (oci). We conducted separate experiments for each language family, training the model with the HRL to English MT task and evaluating it in a zero-shot setting with corresponding ELRLs.

In total, we have 3 HRLs (English, Hindi, and Spanish) and 12 ELRLs. All the test datasets are sourced from FLORES-200 (NLLB Team, 2022), while the hi-en dataset is obtained from AI4Bharat (Ramesh et al., 2022), and the es-en dataset is from Rapp (2021). The development set of FLORES-200 was utilized as a parallel dataset for supervised noise injection. A small amount of monolingual data was used for SELECTNOISE and other baseline methods. Here, we used 1000 examples for each ELRL. Detailed dataset statistics and data sources are presented in Appendix D. In Appendix C, we provide an overview of the lexical similarity between HRLs and ELRLs.

## 3.2 Baselines

We compare the SELECTNOISE model with several strong baselines, including a traditional data augmentation model, lexical similarity-based models, and a model based on random noise injection. Details of each baseline are presented below:

- **Vanilla NMT:** A standard transformer-based NMT model (Vaswani et al., 2017a) with BPE algorithm (Sennrich et al., 2016b).

| Models | Indo-Aryan | | | | | | | | Romance | | | | Average |
|---|---|---|---|---|---|---|---|---|---|---|---|---|---|
| | bho | hne | san | mai | mag | awa | npi | kas | cat | glg | ast | oci | |
| Vanilla NMT | 40.3 | 46.8 | 22.3 | 40.0 | 49.3 | 47.6 | 29.6 | 21.3 | 33.0 | 41.0 | 40.7 | 33.0 | 37.08 |
| Word-drop | 39.5 | 47.2 | 21.8 | 40.6 | 49.0 | 47.6 | 28.6 | 20.6 | 37.6 | 43.6 | 43.4 | 36.0 | 37.96 |
| BPE-drop | 39.1 | 46.8 | 22.6 | 40.4 | 48.7 | 46.7 | 29.2 | 21.1 | 33.8 | 41.7 | 41.5 | 33.0 | 37.05 |
| SwitchOut | 36.1 | 43.2 | 20.1 | 38.2 | 45.6 | 42.7 | 28.3 | 18.8 | 29.0 | 34.9 | 34.9 | 29.1 | 33.41 |
| OBPE | 41.3 | 47.5 | 23.4 | 41.8 | 50.4 | 49.7 | 30.5 | 21.1 | 34.1 | 41.2 | 41.3 | 33.8 | 38.00 |
| BPE-Dropout | 39.8 | 47.4 | 22.5 | 39.9 | 49.6 | 47.7 | 29.3 | 21.2 | 33.2 | 40.8 | 41.4 | 33.0 | 37.15 |
| Random Char Noise | 40.9 | 48.4 | 23.8 | 40.8 | 50.0 | 47.5 | 31.2 | 21.9 | 40.9 | 46.1 | 46.4 | 38.2 | 39.68 |
| SELECTNOISE Model | | | | | | | | | | | | | |
| SELECTNOISE + Greedy | 42.1 | **51.0** | 25.2 | **43.4** | **51.7** | **49.9** | 33.4 | **23.7** | **42.0** | **47.1** | 47.4 | 38.5 | **41.28** |
| SELECTNOISE + Top-k | **42.4** | 49.9 | **26.0** | 43.0 | 51.0 | 48.8 | 33.4 | 23.3 | 41.5 | 47.1 | **47.8** | 38.5 | 41.06 |
| SELECTNOISE + Top-p | 42.0 | 49.6 | 24.1 | 42.4 | 50.6 | 48.8 | **33.6** | 23.3 | 41.6 | 47.1 | 47.5 | **38.8** | 40.78 |
| Supervised Noise Injection Model | | | | | | | | | | | | | |
| Selective noise + Greedy | 41.4 | 49.1 | 25.4 | 42.2 | 50.1 | 48.7 | 32.9 | 22.2 | 41.6 | 47.2 | 47.7 | 38.7 | 40.60 |
| Selective noise + Top-k | 41.7 | 49.3 | 26.3 | 43.3 | 50.8 | 48.7 | 34.2 | 23.6 | 41.9 | 46.8 | 47.5 | 38.7 | 41.10 |
| Selective noise + Top-p | 41.4 | 49.9 | 27.3 | 43.3 | 51.6 | 48.9 | 33.9 | 23.4 | 41.6 | 47.7 | 48.2 | 39.0 | 41.35 |

Table 1: Zero-shot chrF (↑) scores results for ELRLs → English

| Models | Indo-Aryan | | | | | | | | Romance | | | | Average |
|---|---|---|---|---|---|---|---|---|---|---|---|---|---|
| | bho | hne | san | mai | mag | awa | npi | kas | cat | glg | ast | oci | |
| Vanilla NMT | 11.1 | 17.2 | 2.7 | 10.1 | 18.5 | 18.3 | 5.1 | 2.6 | 5.3 | 10.1 | 12.3 | 5.2 | 9.86 |
| Word-drop | 8.7 | 13.7 | 1.9 | 7.7 | 15.2 | 16.1 | 3.0 | 1.6 | 6.9 | 10.7 | 13.3 | 6.5 | 8.76 |
| BPE-drop | 10.8 | 16.1 | 2.7 | 10.0 | 17.2 | 17.8 | 4.0 | 2.1 | 5.1 | 9.1 | 11.2 | 4.7 | 9.23 |
| SwitchOut | 4.3 | 7.7 | 1.4 | 4.9 | 8.4 | 7.9 | 2.9 | 1.2 | 3.5 | 6.3 | 8.2 | 3.8 | 5.04 |
| OBPE | 11.1 | 16.6 | 2.9 | 10.4 | 18.7 | 19.7 | 4.8 | 1.9 | 6.2 | 10.7 | 12.9 | 6.1 | 10.16 |
| BPE-Dropout | 11.6 | 17.5 | 3.1 | 10.1 | 19.3 | 18.3 | 5.4 | 2.5 | 5.4 | 10.1 | 13.0 | 5.4 | 10.14 |
| Random Char Noise | **12.8** | 18.8 | 3.1 | 10.2 | 19.4 | 18.6 | 6.3 | 2.9 | **10.9** | 14.3 | 17.2 | 8.7 | 11.93 |
| SELECTNOISE Model | | | | | | | | | | | | | |
| SELECTNOISE + Greedy | 12.5 | **20.1** | 3.7 | 11.9 | **21.2** | **20.2** | 7.1 | 3.0 | 10.8 | **15.0** | 17.4 | **9.0** | **12.66** |
| SELECTNOISE + Top-k | 12.3 | 19.7 | **3.8** | **12.0** | 20.2 | 19.5 | **7.2** | 2.8 | 10.5 | **15.0** | **17.5** | 8.8 | 12.44 |
| SELECTNOISE + Top-p | 12.7 | 19.5 | 3.8 | 11.9 | 20.3 | 19.6 | 6.7 | **3.2** | 10.7 | 14.8 | 17.1 | 8.9 | 12.43 |
| Supervised Noise Injection Model | | | | | | | | | | | | | |
| Selective noise + Greedy | 13.1 | 19.5 | 4.0 | 11.8 | 19.6 | 19.3 | 6.8 | 2.4 | 10.5 | 15.0 | 17.9 | 8.9 | 12.4 |
| Selective noise + Top-k | 12.7 | 19.1 | 3.9 | 12.2 | 20.1 | 19.3 | 7.0 | 2.9 | 10.8 | 15.0 | 17.4 | 8.9 | 12.44 |
| Selective noise + Top-p | 12.7 | 20.0 | 4.1 | 12.6 | 21.2 | 19.7 | 7.0 | 2.7 | 10.5 | 15.4 | 18.1 | 9.1 | 12.76 |

Table 2: Zero-shot BLEU (↑) scores results for ELRLs → English

| Models | Indo-Aryan | | | | | | | | Romance | | | | Average |
|---|---|---|---|---|---|---|---|---|---|---|---|---|---|
| | bho | hne | san | mai | mag | awa | npi | kas | cat | glg | ast | oci | |
| Vanilla NMT | 0.500 | 0.531 | 0.368 | 0.500 | 0.559 | 0.576 | 0.435 | 0.377 | 0.295 | 0.390 | 0.406 | 0.232 | 0.431 |
| Word-drop | 0.497 | 0.533 | 0.357 | 0.498 | 0.551 | 0.563 | 0.417 | 0.353 | 0.361 | 0.440 | 0.454 | 0.312 | 0.445 |
| BPE-drop | 0.506 | 0.537 | 0.367 | 0.509 | 0.557 | 0.572 | 0.422 | 0.363 | 0.316 | 0.415 | 0.432 | 0.283 | 0.440 |
| SwitchOut | 0.411 | 0.446 | 0.318 | 0.415 | 0.467 | 0.466 | 0.38 | 0.335 | 0.278 | 0.337 | 0.347 | 0.262 | 0.372 |
| OBPE | 0.502 | 0.525 | 0.371 | 0.502 | 0.561 | 0.583 | 0.436 | 0.381 | 0.306 | 0.404 | 0.416 | 0.266 | 0.438 |
| BPE-Dropout | 0.501 | 0.526 | 0.371 | 0.497 | 0.558 | 0.574 | 0.439 | 0.393 | 0.300 | 0.389 | 0.410 | 0.231 | 0.432 |
| Random Char Noise | 0.521 | 0.547 | 0.371 | 0.501 | 0.569 | 0.584 | 0.441 | 0.380 | 0.391 | 0.487 | 0.491 | 0.319 | 0.467 |
| SELECTNOISE Model | | | | | | | | | | | | | |
| SELECTNOISE + Greedy | 0.525 | 0.563 | **0.386** | **0.511** | **0.578** | **0.606** | **0.458** | **0.394** | 0.392 | 0.499 | 0.511 | 0.319 | 0.478 |
| SELECTNOISE + Top-k | 0.524 | 0.558 | **0.386** | 0.507 | 0.576 | 0.599 | 0.454 | 0.388 | **0.400** | 0.497 | **0.516** | **0.321** | 0.477 |
| SELECTNOISE + Top-p | **0.527** | **0.599** | 0.372 | 0.505 | 0.573 | 0.599 | 0.457 | 0.391 | 0.399 | **0.501** | 0.509 | **0.321** | **0.479** |
| Supervised Noise Injection Model | | | | | | | | | | | | | |
| Selective noise + Greedy | 0.527 | 0.560 | 0.389 | 0.507 | 0.572 | 0.600 | 0.451 | 0.381 | 0.392 | 0.499 | 0.511 | 0.319 | 0.476 |
| Selective noise + Top-k | 0.526 | 0.549 | 0.401 | 0.509 | 0.573 | 0.463 | 0.463 | 0.390 | 0.400 | 0.494 | 0.506 | 0.326 | 0.467 |
| Selective noise + Top-p | 0.524 | 0.558 | 0.400 | 0.510 | 0.584 | 0.455 | 0.455 | 0.386 | 0.391 | 0.501 | 0.512 | 0.321 | 0.466 |

Table 3: Zero-shot BLEURT (↑) scores results for ELRLs → English

| Models | Indo-Aryan | | | | | | | | Romance | | | | Average |
|---|---|---|---|---|---|---|---|---|---|---|---|---|---|
| | bho | hne | san | mai | mag | awa | npi | kas | cat | glg | ast | oci | |
| Vanilla NMT | 0.642 | 0.676 | 0.471 | 0.621 | 0.711 | 0.736 | 0.542 | 0.387 | 0.499 | 0.534 | 0.497 | 0.408 | 0.560 |
| Word-drop | 0.659 | 0.702 | 0.494 | **0.650** | 0.725 | 0.747 | 0.564 | 0.409 | 0.484 | 0.551 | 0.538 | 0.421 | 0.579 |
| BPE-drop | 0.653 | 0.687 | 0.497 | 0.645 | 0.711 | 0.732 | 0.554 | 0.408 | 0.438 | 0.515 | 0.505 | 0.389 | 0.560 |
| SwitchOut | 0.565 | 0.605 | 0.462 | 0.564 | 0.626 | 0.632 | 0.533 | 0.394 | 0.405 | 0.461 | 0.445 | 0.362 | 0.504 |
| OBPE | 0.664 | 0.676 | 0.452 | 0.630 | 0.707 | 0.740 | 0.544 | 0.392 | 0.456 | 0.524 | 0.501 | 0.400 | 0.557 |
| BPE-Dropout | 0.644 | 0.672 | 0.471 | 0.616 | 0.710 | 0.733 | 0.537 | 0.381 | 0.503 | 0.534 | 0.500 | 0.411 | 0.559 |
| Random Char Noise | 0.673 | 0.700 | 0.492 | 0.641 | 0.725 | 0.746 | 0.559 | 0.401 | 0.522 | 0.610 | 0.584 | 0.441 | 0.591 |
| SELECTNOISE Model | | | | | | | | | | | | | |
| SELECTNOISE + Greedy | 0.672 | **0.714** | 0.493 | 0.647 | **0.735** | **0.765** | 0.575 | 0.412 | 0.523 | 0.620 | 0.598 | 0.434 | 0.599 |
| SELECTNOISE + Top-k | **0.678** | 0.708 | **0.504** | 0.649 | 0.730 | 0.758 | 0.585 | **0.419** | 0.524 | 0.621 | **0.603** | 0.438 | **0.601** |
| SELECTNOISE + Top-p | 0.677 | 0.559 | 0.502 | 0.643 | 0.730 | 0.758 | **0.586** | 0.411 | **0.526** | **0.625** | 0.600 | **0.442** | 0.588 |
| Supervised Noise Injection Model | | | | | | | | | | | | | |
| Selective noise + Greedy | 0.681 | 0.711 | 0.505 | 0.649 | 0.728 | 0.761 | 0.582 | 0.411 | 0.522 | 0.618 | 0.603 | 0.441 | 0.601 |
| Selective noise + Top-k | 0.677 | 0.700 | 0.506 | 0.655 | 0.703 | 0.757 | 0.581 | 0.414 | 0.522 | 0.623 | 0.605 | 0.439 | 0.598 |
| Selective noise + Top-p | 0.680 | 0.708 | 0.511 | 0.655 | 0.738 | 0.756 | 0.589 | 0.414 | 0.522 | 0.623 | 0.605 | 0.439 | 0.603 |

Table 4: Zero-shot COMET (↑) scores results for ELRLs → English

| Models | Languages | | |
|---|---|---|---|
| | bho | san | npi |
| Annotator set-1 | | | |
| Vanilla NMT | 3.54 | 2.42 | 2.21 |
| BPE Dropout | 3.29 | 2.37 | 1.83 |
| SELECTNOISE Model | **4.17** | **2.83** | **2.50** |
| Annotator set-2 | | | |
| Vanilla NMT | 3.42 | 1.96 | 2.17 |
| BPE Dropout | 2.79 | 1.83 | 1.96 |
| SELECTNOISE Model | **3.54** | **2.17** | **2.21** |

Table 5: Human evaluation results - Average score

- **Word-drop (Sennrich et al., 2016a):** In this baseline, 10% words embeddings from each sentence of the source HRL is set to zero. This is a data augmentation technique to create training data ELRLs. The rest of the steps are similar to the SELECTNOISE model.

- **BPE-drop:** This approach is similar to the word-drop baseline but uses BPE tokens instead of words.

- **SwitchOut (Wang et al., 2018):** In this baseline, we randomly replace 10% of the source and target words with randomly sampled words from their respective vocabularies. We use the officially released implementation.

- **OBPE (Patil et al., 2022):** OBPE modifies the learned BPE vocabulary to incorporate overlapping tokens from both HRL and LRLs, even if the token is not frequent. We utilized the official implementation.

- **BPE Dropout (Provilkov et al., 2020):** It is based on the BPE algorithm to learn the vocabulary and generates non-deterministic segmentations for input text *on-the-fly* during training. We use a dropout value of 0.1.

- **Random Char Noise (Aepli and Sennrich, 2022):** This baseline methodology is similar to the proposed SELECTNOISE approach; but, noise injections are done randomly.

### 3.3 Evaluation Metrics

All the model performances are compared using both automated and human evaluation metrics. In line with recent research on MT for LRLs, we employ two types of automated evaluation metrics (NLLB Team, 2022; Siddhant et al., 2022). Specifically, lexical match-based metrics: BLEU (Papineni et al., 2002) and chrF (Popović, 2015) and learning-based metrics: BLEURT (Sellam et al., 2020) and COMET (Pu et al., 2021).

We further conducted the human evaluation to ensure the reliability of the performance gain. Three languages from the Indo-Aryan family ( Bhojpuri, Nepali, and Sanskrit) were selected based on their high, moderate, and low lexical similarity with the HRL (Hindi). To manage the annotators' workload effectively, we limited our evaluation to three models: Vanilla NMT, BPE Dropout, and SELECTNOISE. For each language, the human evaluation set consisted of 24 examples, and translations were obtained from above mentioned three models. Two annotators were employed for each language to ensure the inter-annotator agreement, and two ratings were obtained for each example from these annotators. All annotators held at least a master's degree, were native speakers of the respective language and demonstrated proficiency in English. We use *Crosslingual Semantic Text Similarity (XSTS)* metric (Agirre et al., 2012), which is widely adopted in the MT research for human evaluation. The XSTS metric employs a 1-5 evaluation scale, where 1 represents a very bad translation and 5 represents a very good translation.

## 4 Results and Discussions

In this section, we will discuss results, observations and findings. The zero-shot automated evaluation scores are presented in Tables 1, 2, 3 and 4. The results are reported with greedy, top k (k = 50), and top-p (p = 0.25) sampling strategies. Table 5 presents the human evaluation results.

**SELECTNOISE vs. Baselines:** The proposed and other models that incorporate lexical similarity have demonstrated superior performance compared to the Vanilla NMT model. While general data augmentation techniques like Word-drop and SwitchOut exhibit performance similar to the Vanilla NMT model, they perform poorly when compared to OBPE and BPE-Dropout models. These results indicate the importance of considering monolingual data from ELRLs in the modeling However, random noise injection and the SELECTNOISE approach outperform the OBPE and BPE-Dropout models, indicating the effectiveness of noise injection-based modeling techniques. In conclusion, the careful selection of noise candidates, as done in the SELECTNOISE approach, has outperformed the random noise model (second best) and emerged as the state-of-the-art model.

**Selective vs. Random Noise injection:** Unsupervised selective noise injection approaches exhibit a

larger performance gain compared to the random noise injection model. This observation emphasizes the importance of a systematic selective candidate extraction and noise injection process.

**Lexical vs. Learned Evaluation Metrics:** We observe a strong correlation between lexical match-based metrics, such as BLEU and chrF scores. Further, semantic-based metrics like BLEURT and COMET exhibit similar trends to lexical match metrics, indicating a high level of correlation. This emphasizes the reliability of evaluation scores.

**Automated vs. Human Evaluation:** The proposed SELECTNOISE model outperforms both baselines in human evaluation across all three languages. The model demonstrates acceptable zero-shot performance for ELRLs, with a strong correlation with automated evaluation scores.

**Performance across Language Families:** Unsupervised selective noise injection consistently outperforms all the baselines across ELRLs, with few exceptions. The model exhibits similar performance trends across both language families.

**Unsupervised vs. Supervised Noise Injection:** The unsupervised SELECTNOISE model performs comparably to the supervised model, with slight variations depending on the language and family. The performance gap between the two models is minimal, indicating their equal strength.

**Performance vs. Sampling Strategies:** The performance with different sampling techniques is compared, and it is observed that the greedy approach for SELECTNOISE performs better for the majority of languages. This finding indicates the existence of one-to-one lexical mapping across HRL and ELRLs. However, other sampling approaches are also effective for a subset of ELRLs.

**Overall Performance:** As we can observe from the average automated evaluation scores, the proposed SELECTNOISE model outperforms all the baselines by a significant margin. It also exhibits comparable performance to the supervised model, and this performance persists across different language families. These findings satisfy our hypothesis, leading us to conclude that the proposed SELECTNOISE model is a state-of-the-art model for English-to-ELRLs MT systems.

## 5 Further Analyses

In this section, we perform a detailed analysis with SELECTNOISE to understand factors contributing to performance gain and analyze robustness.

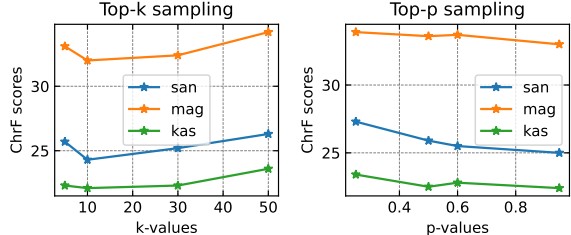

Figure 4: Proposed model performance trends with various k and p values from top-k and top-p sampling respectively.

| Language | Data size | BLEU | chrF |
|---|---|---|---|
| **hne** | 997 | 19.5 | 49.6 |
| | 6000 | **20.3** | **50.3** |
| **mai** | 997 | 11.9 | 42.4 |
| | 6000 | **12.4** | **43.2** |
| **npi** | 997 | 6.7 | 33.6 |
| | 6000 | **7.2** | **33.8** |

Table 6: Model performance with larger monolingual data

**Performance Trend with Top-k and Top-p:** In Figure 4, the performance trend of the proposed model with varying values of k and p for top-p and top-k sampling is depicted. The candidate pool consists of a maximum of 61 characters (a range for k-value selection). The model performs best with a k-value of 50 and a p-value of 0.25, offering valuable insights for optimizing its performance through parameter selection.

**Impact of Monolingual data size:** The proposed SELECTNOISE model relies on the small monolingual dataset of ELRLs. We investigate the impact of a large monolingual dataset on the model's performance for ELRLs. Table 6 demonstrates that a larger dataset leads to a performance boost, suggesting the extraction of more meaningful noise injection candidates.

**Language similarity Vs. Performance:** Figure 5 illustrates the comparative trend of lexical similarity score between ELRLs and HRLs and performance (ChrF score). It can be observed that lexically similar languages boost the model's performance, leading to an improved cross-lingual transfer for the SELECTNOISE model. For example, languages like Kashmiri (kas), which have the lowest similarity, exhibit the lowest performance, whereas Chhattisgarhi, with the highest lexical similarity, demonstrates the highest performance.

**Performance with Less related Languages:** We evaluate the zero-shot translation performance of Vanilla NMT and proposed SELECTNOISE models with two relatively less lexically similar ELRLs. These two languages belong to distinct language families, namely Bodo (Sino-Tibetan) and Tamil

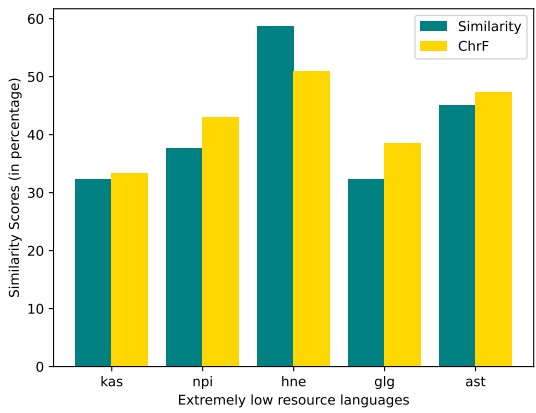

Figure 5: Language similarity vs. Performance.

| Language | Model | BLEU | chrF |
|----------|-------|------|------|
| Bodo | Vanilla NMT | 2.4 | 18.2 |
| | SELECTNOISE | **2.7** | **18.7** |
| Tamil | Vanilla NMT | 0.6 | 11.7 |
| | SELECTNOISE | **0.9** | **13.3** |

Table 7: Zero-shot translation performance of Vanilla NMT vs. SELECTNOISE on less related LRLs with HRL (Hindi)

(Dravidian). Bodo has Devanagari script, while Tamil employs script conversion to match HRL (Hindi) script. The results are reported in Table 7. It is observed that the performance gain is minimal due to the dissimilarity of ELRLs with the corresponding HRL.

## 6 Related Work

**MT for Low-resource Languages:** Limited parallel corpora for many LRLs lead to active research in multilingual MT. These are grounded with transfer learning to enable cross-lingual transfer (Nguyen and Chiang, 2017; Zoph et al., 2016) and allow related languages to learn from each other (Fan et al., 2021; Costa-jussà et al., 2022; Siddhant et al., 2022). Further, these approaches can be extended by grouping training data based on relatedness (Neubig and Hu, 2018) or clustering similar languages (Tan et al., 2019) to improve performance for LRLs. In another direction, monolingual corpora are combined with parallel corpora to enhance translation quality (Currey et al., 2017) or used for unsupervised NMT (Artetxe et al., 2018; Lewis et al., 2020), reducing the reliance on parallel data. Back-translated data is also widely utilized for training MT systems for LRLs (Sugiyama and Yoshinaga, 2019; Edunov et al., 2018). More recently, models powered by large multilingual pre-trained language models (mLLMs) enable MT with limited language resources (NLLB Team, 2022; Zhu et al., 2023). These models have shown ac-

ceptable performance for many LRLs. However, adapting these models to ELRLs is challenging because they often lack parallel data, have limited monolingual data, and are absent from mPLMs. In contrast, we propose a model that only requires a small monolingual data (1000 examples).

**Data Augmentation for Low-resource MT:** The limited availability of parallel data leads to a wide range of data augmentation approaches (Zhang et al., 2019; Gao et al., 2019; Cotterell and Kreutzer, 2018). Traditional approaches include perturbation at the word level, such as word dropout (Sennrich et al., 2016a), word replacement (Wang et al., 2018) and soft-decoupling (SDE; Wang et al. (2019)) to improve the cross-lingual transfer for LRLs. Such perturbation acts as a regularizer and enhances robustness to spelling variations; however, their impact is limited (Aepli and Sennrich, 2022). In a different research direction, noise injection-based modeling (Sperber et al., 2017; Karpukhin et al., 2019) has been explored to test the robustness of MT systems. More recently, lexical match-based models have been explored to improve the cross-lingual transfer by vocabulary overlapping (Patil et al., 2022), non-deterministic segmentation (Provilkov et al., 2020) and noise injection (Aepli and Sennrich, 2022; Blaschke et al., 2023). In contrast to these methods, we propose a linguistically inspired systematic noise injection approach for ELRLs.

## 7 Conclusion

This study presents an effective unsupervised approach, SELECTNOISE, for cross-lingual transfer from HRLs to closely related ELRLs through systematic character noise injection. The approach involves extracting selective noise injection candidates using BPE merge operations and edit operations. Furthermore, different sampling techniques are explored during the noise injection to ensure diverse candidate sampling. The model required only a small (1K example) monolingual data in ELRLs. The proposed model consistently outperformed strong baselines across 12 ELRLs from two diverse language families in the ELRLs-to-English MT task. The cumulative gain is 11.3% (chrF) over vanilla NMT. Furthermore, the model demonstrated comparative performance to a supervised noise injection model. In the future, we will extend SELECTNOISE to English-to-ELRL MT task, as well as other NLG tasks and languages.

## Limitations

The present study is based on the assumption of closely related high-resource languages and extremely low-resource languages to facilitate improved cross-lingual transfer. However, the proposed method may not be effective for ELRLs that have a different script and lack a script conversion or transliteration tool. Additionally, the model's performance may be suboptimal for languages that share the same script but exhibit significant lexical differences, such as Hindi and Bodo. Furthermore, this study focuses on ELRLs to English translation, and it remains to be explored whether the noise injection approach is beneficial for the English to ELRLs translation task.

## Acknowledgements

We extend our heartfelt gratitude to all human evaluators whose meticulous assessment of the translation performance of the proposed model and various baselines played a pivotal role in the success of this research. Furthermore, we wish to express our deep appreciation to the anonymous reviewers and the esteemed meta-reviewer for their invaluable and insightful suggestions, which greatly improve the quality of the manuscript.

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

## A  Character Candidate Pool ($\mathcal{S}_c$)

A sample structure of character candidate pool $\mathcal{S}_c$ is illustrated in Fig. 6. More details are presented in Section 2.1.1.

## B  Performance for HRLs

Table 8 analyzes the performance of the proposed model for HRLs across both language families. It demonstrates comparable performance with the vanilla NMT model for HRLs while boosting the performance of ELRLs. This highlights the effectiveness of the proposed model in handling both HRLs and ELRLs.

| Models | Languages | | | |
|---|---|---|---|---|
| | BLEU | chrF | BLEURT | COMET |
| Hindi (hi) | | | | |
| Vanilla NMT | 33.4 | 60.2 | 0.724 | 0.868 |
| Random Char Noise | 33.0 | 59.5 | 0.722 | 0.865 |
| SELECTNOISE | **34.2** | **60.5** | **0.726** | **0.869** |
| Spanish (es) | | | | |
| Vanilla NMT | 21.5 | 53.5 | 0.695 | 0.810 |
| Random Char Noise | 21.7 | 53.0 | 0.689 | 0.806 |
| SELECTNOISE | **21.3** | **53.1** | **0.689** | **0.869** |

Table 8: Comparative performance for HRLs across both Indo-Aryan and Romance families.

Candidate Pool ($S_c$) Template
```
{
    C₁: {'I': f₁, 'D': f₃, 'S': { E₁: f₄, E₂: f₅, … }},
    C₂: {'I': f₄, 'D': f₃, 'S': { E₃: f₆, E₂: f₇,… }},
        ⋮
    Cᵢ: {'I': f₂, 'D': f₄, 'S': { E₁: f₂, E₃: f₁, … }}
        ⋮
    Cₙ: {'I': f₁, 'D': f₄, 'S': { E₁: f₄, E3: f₅, … }}
}
```

Few Sample Elements of Candidate Pool ($S_c$)
```
{
 'ॢ':{'I':62,'D':1561,'S':{'ह': 1482,…}}
 'र':{'I':92,'D':97,'S':{'ा':1482,…}}
 'ि':{'I':1552,'D':15,'S':{'य':397,…}}
 'S': {'I': 0,'D': 33, 'S': {}}
}
```

Figure 6: *Top* is a template for the character candidate pool $\mathbf{S}_c$. The operations are: *insertion* ($I$), *deletion* ($D$) and *substitution* ($S$). The character $c_i$ represents the $i^{th}$ element of $\mathcal{S}_c$, which is an HRL character, $c'_1 \ldots c'_k$ denote the corresponding substituting candidates from ELRLs and $f$ is the associated frequencies. The *Bottom* shows a few sample elements of the $\mathbf{S}_c$.

## C HRL-ELRL lexical similarity measurement

Figure 7 shows the lexical similarity between HRL and related ELRLs for both language families. Lexical similarity is obtained using the longest common subsequence (LCS; Melamed (1995)).

## D Datasets

Detailed statistics of datasets used in our experiments are shown in Table 9. For performing analysis on less-related language, we use the general test set of IndicTrans2 (AI4Bharat et al., 2023) for Bodo and FLORES200 test set for Tamil. For

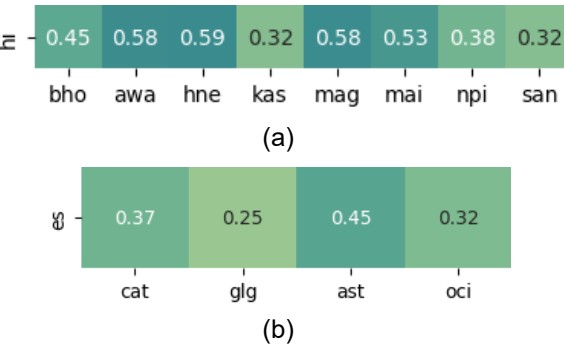

Figure 7: Lexical similarity heatmap between HRL and its related ELRLs. Fig. (a) depicts a similarity score for the Indo-Aryan family where HRL is Hindi. Fig. (b) depicts a similarity score for the Romance family where HRL is Spanish. *Note: Darker color denotes more similarity.*

OBPE baseline, we use a dev set of FLORES-200 consisting of 997 as a monolingual corpus for learning the overlap vocabulary.

## E Implementation Details

Our vanilla NMT model is based on standard transformer architecture consisting of 6 encoder and decoder layers. We trained our model for a maximum epoch of 15. We use Adam (Kingma and Ba, 2015) optimizer with $\beta_1 = 0.9$ and $\beta_2 = 0.98$. We set a learning rate of 0.0005. We use a dropout of 0.2. We performed data normalization and preprocessing using IndicNLP library[4]. We perform our experiments using fairseq[5] library. For evaluation we use the lexical match-based BLEU metric[6] (Pa-

---

[4] https://github.com/anoopkunchukuttan/indic_nlp_library
[5] https://github.com/pytorch/fairseq
[6] nrefs:1|case:mixed|eff:no|tok:13a|smooth:exp|version:2.3.1

| Languages | #Dev | #Test | HRL | Source |
|---|---|---|---|---|
| bho, mag, mai, npi, awa, san, kas, hne | 997 | 1012 | hi | FLORES-200 |
| ast, cat, glg, oci | 997 | 1012 | es | FLORES-200 |

| HR Lang Pair | #Train | # Dev | #Test | Source |
|---|---|---|---|---|
| hi-en | 10.1M | 997 | 1012 | AI4Bharat |
| es-en | 6.6M | 997 | 1012 | Rapp (2021) |

Table 9: Dataset statistics and language details

pineni et al., 2002), chrF[7] (Popović, 2015) metric, semantic-based BLEURT[8] (Sellam et al., 2020), and COMET[9] (Pu et al., 2021) metrics.

# F  Sample Translations

Fig. 8 presents sample translations from Random Character Noise, SELECTNOISE and Supervised Character Noise injection models.

---

[7] nrefs:1|case:mixed|eff:yes|nc:6|nw:0|space:no|version:2.3.1
[8] Reported using BLEURT20 checkpoint
[9] Reported using wmt22-comet-da model

**BHO:** ट्रंप के हई घोसना, तुर्की के राष्ट्रपति रेसेप तइप एर्डोअन से उनकर फ़ोन पर बातचीत के बाद आइल बा.

**ENG:** The announcement was made after Trump had a phone conversation with Turkish President Recep Tayyip Erdoğan.

**RCN:** Trump opened up about the phone call with Turkish President Recep Tayyip Erdogan.

**UCN:** After a phone call with Turkish President Recep Tayyip Erdogan, Trump announced his decision.

**SCN:** Trump's announcement came after a phone call with Turkish President Recep Tayyip Erdogan.

**HNE:** पुलिस ह बताए कि शव करीब इक दिन से पड़े हुए लगत हे।

**ENG:** Police said that the body appeared to have been there for about a day.

**RCN:** The police said the body had been lying on the ground for the past few days.

**UCN:** The police said the bodies had been lying on the road for the past few days.

**SCN:** The police said the body had been lying for the past several days.

**GLG:** A ONU tamén ten previsto crear un fondo para axudar aos países afectados polo quentamento global a afrontar o seu impacto.

**ENG:** The UN also intends to set up a fund to relieve the countries affected by the global fear of facing or having a serious impact.

**RCN:** The UN is also planning to set up a fund to tax years of countries affected by global pole to cope with or have an impact.

**UCN:** The UN also intends to set up a fund to relieve the countries affected by the global fear of facing or having a serious impact.

**SCN:** The UN also plans to set up a fund to relieve the countries affected by the global warming to face up to their impact.

**AST**: Munchos cudadanos de Bishkek acusaron a los manifestantes sureños del desorde.

**ENG:** Several Bishkek residents blamed protesters from the south for the lawlessness.

**RCN:** Cubans of Bishkek accused southern protestors of the deorde.

**UCN:** Many people in Bishkek accused the Swiss demonstrators of disorder.

**SCN:** Many people in Bishkek accused southern demonstrators of disorder.

Figure 8: Sample translations from various models for ELRLs to English MT direction. RCN: Random Character Noise injection model, UCN: Unsupervised Character Noise injection model (i.e., SELECTNOISE model) and SCN: Supervised Character Noise injection model.