# OpenReview forum: "$\textit{SelectNoise:}$ Unsupervised Noise Injection to Enable Zero-Shot Machine Translation for Extremely Low-resource Languages"
_EMNLP/2023/Conference — EMNLP 2023 Findings_

### Official Review · Reviewer_2AWf · 2023-07-30

**Soundness:** 4

**Excitement:**

2: Mediocre: This paper makes marginal contributions (vs non-contemporaneous work), so I would rather not see it in the conference.

**Paper Topic And Main Contributions:**

The paper proposes a method to synthesize a pseudo parallel corpus for a low resource language (LRL) pair from a parallel corpus of a linguistically-close high resource language (HRL) pair. The method first build a database of HRL character edit operations which are likely to produce LRL (sub)words from HRL words in an unsupervised fashion. Then characters at random positions in the HRL parallel corpus are changed by a character edit operation selected from the database.

The experimental results show translation quality improvements consistently across serval languages against various vocabulary regularization techniques.

**Questions For The Authors:**

A. Line 221-223: I imagine some HRL characters are associated more with some particular edit operation and less with others. Did you test biasing edit operation sampling depending on the HRL character?

B. Line 267-269: Did you compute the softmax with raw counts? For example, if the candidate characters are "a" and "b", and they were observed 5 and 2 times respectively, their selection probabilities are exp(5) / (exp(5) + exp(2)) and exp(2) / (exp(5) + exp(2))? if so, frequent characters are more likely to be sampled. (exp(5) / exp(2)  ~ 20, 5 / 2 = 2.5)

C. Line 386: "Four languages ... namely Bhojpuri, Nepali and Sanskrit ..." What's the forth language?

D. Line 389-394: "Each language's evaluation set comprised 24 examples, which were individually assessed by a single annotator. ... each evaluation set was evaluated by two annotators." I'm confused. How many human ratings were annotated to each example?


**Reasons To Accept:**

* The proposed method doesn't require parallel corpora thus can be applicable to many low resource language pairs.

* The proposed method is linguistically intuitive and has a potential to incorporate more linguistic knowledge.

**Reasons To Reject:**

* The proposed method is expected to work better with LRLs whose linguistic structures are close to the base HRL, but less with other LRLs. Though the authors conducted experiments with many LRLs, the paper didn't provide detailed analyses. It is difficult to speculate how these findings will generalize into other language families.

* As the authors acknowledged in Limitations, the proposed method is applicable only when the LRL has a HRL which is close not only in linguistic features but also scripts. The paper may not be appealing to a wide audience.

**Reproducibility:**

4: Could mostly reproduce the results, but there may be some variation because of sample variance or minor variations in their interpretation of the protocol or method.

**Reviewer Confidence:**

5: Positive that my evaluation is correct. I read the paper very carefully and I am very familiar with related work.

**Typos Grammar Style And Presentation Improvements:**

* Since the comparison to supervised noise injection is a part of the paper's main claim, Algorithm 2 should be included in the main part.

* Some people might be confused by calling  the experiment setup zero-shot. My understanding is that this paper proposes a method to synthesize a pseudo ELRL-English parallel corpus from a HRL-English parallel corpus. With this framing, the experiment is not zero-shot.

---

> ### Author Rebuttal · Authors · 2023-08-29
>
> We thank the reviewer for the constructive feedback, and phrases like ‘method is linguistically intuitive’ are encouraging. In the final version of the paper, we will address missing citations, grammatical and spelling errors, and all presentation issues.
>
> **On the point of "paper didn't provide detailed analyses":**  The primary hypothesis we are operating is to enable MT technology for lexically similar ELRLs. In fact, the proposed model boosts the performance for less similar languages like San and Kas (from Table 1-4), and Bodo and Tamil (Table 7) languages. **On generalization capabilities, we have experimented with 12 ELRLs (Table 1-4) and compared them with different baseline models with four evaluation metrics, human evaluation (Table 5), experiments with different sampling algorithms, supervised experiments, performance for HRLs (Table 8 in the appendix), analysis with top-p and top-k values (Figure 2), analysis with different-sized monolingual data (Table 6), and performance analysis for less related languages (Table 7).** All these experiments and analyses provide evidence that the model generalizes across two language families and 12 ELRLs/dialects. Moreover, the proposed novel noise injection model based on BPE merge operations, edit distance, and sampling techniques is an automated process.  This can be applied to any ELRLs and language families as long as they have a small dataset (1000) of monolingual examples. With these points, we conclude that the proposed model's generalization capabilities can be easily scaled to any closely related ELRLs and HRLs.
>
> **On the point of "The paper may not be appealing to a wide audience":** The proposed model aims to advance and scale language coverage for MT research and present a robust foundational model for researchers for ELRLs. Although the proposed model requires similar scripts, there are many ELRLs that have related HRLs in the same script. Additionally, script conversion stands as a plausible alternative. With this work, our broader goal is to enable language technology for a general audience. As per Ruder (2022), there are around 7,000 languages spoken across the globe, and 400 languages have more than 1 million speakers, with approximately 1,200 languages having more than 100,000 speakers. Despite this, only around 100 languages are incorporated into large pre-trained models, and a few hundred have MT technology enabled. Furthermore, in ACL 2008, it was found that 63% of all papers focused on English. A recent similar study from ACL 2021 concluded that almost 70% of papers were only evaluated using English. Even a decade later, there has been little change. Considering these motivations, our paper is positioned as a step towards enabling MT technology for larger languages with a significant number of speakers (listed in the table below) that may interest the broader audience. Here are the number of speakers for considered languages (source: wikipedia):
>
> | Languages | # of speakers |
> | ---------------------------- | --------------- |
> | Bhojpuri (bho) | 50M |
> | Chhattisgarhi (hne) | 16M |
> | Magahi (mag) | 12M |
> | Maithili (mai) | 13M |
> | Awadhi (awa) | 3.8M |
> | Nepali (npi) | 2.9M |
> | Kashmiri (kas) | 7.1M |
> | Sanskrit (san) | 24, 821 |
> | Bodo (brx) | 1.4M |
> | Tamil (tam) | 69M |
> | Catalan (cat) | 9.2M |
> | Asturian (ast) | 5.5M |
> | Galician (glg) | 2.4M |
> | Occitan (oct) | 100,000 - 800,000 |
>
> **Response to  Question A:** We agree with the reviewer's observation on varying HRL-ELRL character associations. In our current setup, we've extracted edit operations with frequencies for multiple ELRLs. For a given HRL character “h” undergoing three operations, each with “m” character mappings across “n” ELRLs, we have 3xmxn options with different frequencies. Considering any explicit association (heuristics/rules) in MT training may encourage biases for some ELRLs. Moreover, this limits the proposed model's scalability to other language families. To tackle the imbalanced data operations associate, we introduce divergence by considering edit operations with equal probabilities and using sampling decoding techniques to add diversity. While our framework indirectly considers associations, automated explicit modeling of such associations is left as future work.
>
> **Response to  Question B:** Yes, the softmax computation employs the raw counts. In order to foster diversity and mitigate the tendency to sample the most frequent characters, we incorporate both top-k and top-p strategies in addition to the greedy approach.
>
> **Response to  Question C:**  It is a typo; we used three languages for human evaluation: Bhojpuri, Nepali, and Sanskrit.
>
> **Response to  Question D:** We apologize for the confusing language. For human evaluation, we have considered three languages: Bhojpuri, Nepali, and Sanskrit. For each language, we randomly sampled 24 examples for human evaluation. For each example, we obtained two ratings from two different annotators to ensure inter-annotator agreement. We have reported the average scores of each annotator in Table 5.
>
> **On the point of "Model is Zero-shot":** Similar question is raised by reviewer #2. We kindly request you to refer our response to reviewer #2.

---

### Official Review · Reviewer_eEMd · 2023-08-04

**Soundness:** 3

**Excitement:**

3: Ambivalent: It has merits (e.g., it reports state-of-the-art results, the idea is nice), but there are key weaknesses (e.g., it describes incremental work), and it can significantly benefit from another round of revision. However, I won't object to accepting it if my co-reviewers champion it.

**Paper Topic And Main Contributions:**

In this paper, the authors aim to address the translation between extremely low-resource languages (ELRLs) and English. And the authors propose an unsupervised method by utilizing the shared char-level vocabulary to conduct pseudo-parallel sentence pairs for ELRLs.
However, I don't think this simple data-augmentation method is novel in 2023, and the authors only evaluate the translation performance between ELRLs and English, which is inadequate to be the zero-shot translation.

**Reasons To Accept:**

The authors try to address the translation between extremely low-resource languages (ELRLs) and English, and they apply char-level candidates' selective and noise injection strategies to conduct pseudo-parallel sentence pairs for ELRLs.
This method performs better than word-substitute methods.

**Reasons To Reject:**

Although this method performs better than other word-substitute methods, some issues exist as follows:
1. Few languages or language families are involved in experiments, does the method have the generalization on other low-resource languages or language families, and the parameters of ratio in this method are also suited?
2. What are the advantages of this method, compared to the back-translation or translationese method on monolingual data?
3. How to define zero-shot translation and few-shot translation.
4. How to address multi-sense words in different contexts?

**Reproducibility:**

4: Could mostly reproduce the results, but there may be some variation because of sample variance or minor variations in their interpretation of the protocol or method.

**Reviewer Confidence:**

4: Quite sure. I tried to check the important points carefully. It's unlikely, though conceivable, that I missed something that should affect my ratings.

---

> ### Author Rebuttal · Authors · 2023-08-29
>
> We thank the reviewer for the constructive feedback. In the final version of the paper, we will address missing citations, grammatical and spelling errors, and all presentation issues.
>
> **On the point of "limited novelty":** To the best of our knowledge this is the first effort towards systematic unsupervised noise augmentation to enable zero-shot machine translation for ELRLs.  The novel contributing aspects through this work have not been seen in previous work: **(a) we proposed a novel approach based on BPE merge, edit operations and sampling techniques to find noise injection candidates which required only small (1000) monolingual examples for ELRLs. This modeling is automated, linguistically intuitive and scalable.  (b) We show that noise augmentation helps cross-lingual transfer in the NLG task like MT in extremely data-constrained scenarios. Previous work shows benefits for NLU tasks only (Aepli& Sennrich, 2022). (c) Our proposed  noise injection not only shows improvements in transfer among dialects but also between less-related languages like SAN and KAS (see Table 2). This opens up possibilities for further work in noise augmentation for cross-lingual transfer (for instance one-many translation scenarios, other NLG tasks, MT between related languages).** Different data augmentation methods have different motivations, and we systematically investigate transfer between related languages in low-resource settings as our motivation. Existing data augmentation techniques for the MT task mentioned in Section 2 have one or more of the following limitations: (1) may not aim for the zero-shot cross-lingual transfer, (2) may not aim for the MT task and (3) may require (small) parallel in ELRLs, (4) do not utilize lexical similarity. The proposed model focuses on the ELRL -> En MT task in a zero-shot setting and does not require any parallel for ELRLs. Nevertheless, we have considered multiple data augmentation-based state-of-the-art models as baseline models and compared the performance in Table 1-4. The proposed model outperformed all the baselines.
>
> **On the "inclusion of a few ELRLs":** Similar concerns have been raised by Reviewer #1. We kindly request you to refer to the response to Reviewer #1. We are unsure about what the reviewer is referring to with the phrase “the parameters of ratio in this method.” Assuming it pertains to the range of noise injection percentage, we would like to clarify that we have conducted a comprehensive set of ablation studies using different noise ranges. As a result, we have concluded that a noise injection percentage of 5-10% works best.
>
> **On the "comparison with the back-translation or translations models":** The proposed unsupervised selective noise injection does not require any parallel data and only 1000 monolingual data in ELRLs to enable zero-shot translation for ELRLs to En. For ELRLs, neither back-translated nor Translationese-based MT models can be built, as they require an MT system in ELRLs which is primarily the goal of this work: to enable an MT system for ELRLs. The assumption of an existing MT system in ELRLs can not be made.
>
> **On the "definition of Zero-shot and few-shot":** The proposed model only focuses on the zero-shot setting, not in a few-shot setting. Here, the Zero-shot translation (MT) is defined as the training of an MT model without any human-annotated parallel data. If the training of an MT system requires a few human-annotated parallel examples (typically 100 - 1000), it is referred to as a few-shot translation. In the proposed approach, we assume the absence of a human-annotated parallel dataset for ELRLs. Furthermore, by introducing noise into the HRLs human-annotated parallel, the modified parallel dataset acts as a pseudo-parallel dataset for multiple ELRLs, which is used to train the MT model from ELRLs to English. This pseudo-parallel dataset does not necessitate human annotation. As a result, the model we propose is referred to as a zero-shot translation model for ELRLs.
>
> **On the point of "multi-sense words":** Any research is incremental, so we aim to develop an MT system for ELRLs with this work. We assume that many instances of multi-sense words are present in the dataset, and a model trained with this dataset will leverage the broader context and learning to differentiate such words and provide appropriate translations. However, we have not focused on this or not conducted any analysis. This task can be taken separately as future work.

---

### Official Review · Reviewer_4tcT · 2023-08-05

**Typos Grammar Style And Presentation Improvements:** N/A
**Soundness:** 4

**Excitement:**

3: Ambivalent: It has merits (e.g., it reports state-of-the-art results, the idea is nice), but there are key weaknesses (e.g., it describes incremental work), and it can significantly benefit from another round of revision. However, I won't object to accepting it if my co-reviewers champion it.

**Missing References:**

N/A

**Paper Topic And Main Contributions:**

This paper investigates the possibility of building MT system from extremely low-resource languages to English.


----After the rebuttal----

Thanks for the feedback from the authors!
By referring to the newly updated QE results, I believe that some of my concerns are well dispelled, and the effectiveness of the proposed method are strengthed. The authors can add those results into the revision for better claims.

I'll raise my judgement on the soundness of this paper from 3 to 4.

**Questions For The Authors:**

1. Have the authors consider other languages like Tibetan? I'd love to see more ELRLs languages.
2. As the available metrics might not be trained on those involved ELRLs, I think you can provide more QE/metric results (e.g., COMETKIWI, UniTE) to support the significance of the improvements on model performance. You can refer to the official report of WMT Metrics/QE papers to find some methods.

**Reasons To Accept:**

1. Proposed method relieve the reliance on available training resources.
2. Well-written and easy-to-follow.

**Reasons To Reject:**

1. Involved ELRLs might be less to derive convincing conclusion. More languages (especially different families or character-systems) are welcomed to help improve the manuscript. (See Q1)
2. The used metrics might be insufficient as they might have little (or even contradictory) support on the performance measurements (See Q2)

**Reproducibility:**

3: Could reproduce the results with some difficulty. The settings of parameters are underspecified or subjectively determined; the training/evaluation data are not widely available.

**Reviewer Confidence:**

3: Pretty sure, but there's a chance I missed something. Although I have a good feel for this area in general, I did not carefully check the paper's details, e.g., the math, experimental design, or novelty.

---

> ### Author Rebuttal · Authors · 2023-08-29
>
> We thank the reviewer for the constructive feedback, and phrases like ’Well-written and easy-to-follow’' are encouraging. In the final version of the paper, we will address missing citations, grammatical and spelling errors, and all presentation issues.
>
> **On the "less number of ELRLs":**  We have used 12 ELRLs from 2 typologically diverse families (Indo-Aryan and Romance) to evaluate the performance of the proposed model (refer to section 3.1). Additionally, 2 LRLs (Bodo and Tamil) have been employed for model analysis from the Sino-Tibetan and Dravidian language families, resulting in a total count of 14 ELRLs across 4 language families. **The decision regarding the number of ELRL languages was influenced by recent literature and the availability of languages in the FLORES-200 test set. For example, Aepli, N., & Sennrich, R. (2021) employed 5 LRLs; Patil et al. (2022) used 12; Li, Zhaocong, et al. (2022) worked with 5; Khemchandani, Yash, et al. (2021) used 3; Kumar, Sachin, et al. (2021) worked with 7; Lin, Zehui, et al. (2021) used 4 languages, among others.** The proposed model consistently outperforms the best baseline for most of the languages, with an absolute average difference of 1.6 in chrF scores, providing evidence for the model's performance (see Tables 1, 2, 3, and 4 for detailed results). Additionally, in Table 5, we present human evaluations that further support and align with the findings from automated evaluation scores (which correlate with each other). Taking these into consideration, we believe that the chosen number of ELRLs is reasonable. The evaluation involving both automated and human scores adds credibility and lends trustworthy evidence of the model's improvement. In the future, we will explore with Tibetan and other ELRLs.
>
> **On the point of "metrics might be insufficient":** We have considered four widely used automatic evaluation metrics to assess various model performances. Specifically, we employ both metrics based on lexical overlap (chrF and BLEU) and metrics focused on semantics (BLEURT and COMET). The task addressed in this study involves translating from ELRLs to English, where the evaluation scores are reported by comparing the generated English text with Ground Truth English text across all evaluation metrics. The COMET metric also utilizes the source sentence (ELRLs). While COMET might not have been trained with ELRLs, it certainly includes related HRLs, resulting in reliable scores. Moreover, these metrics have generally demonstrated reliable performance in evaluating English sentences, as indicated in the previous literature. Additionally, we have conducted human evaluations to further enhance the reliability of the proposed model. Therefore, evaluating English text using metrics based on lexical overlap, metrics focusing on semantics, and human evaluation collectively contribute to the credibility of the proposed experimental and evaluation framework. Nonetheless, we have computed evaluation scores for both COMETKIWI and UniTE. Due to limitations in readability, we have included COMETKIWI scores here. UniTE exhibits similar trends and correlates with the other metrics.
>
> COMETKIWI scores for Indo-Aryan Family
> | Models | bho | hne | san | mai |mag |awa | npi | kas |
> | -------- | ----- | ---- | ----- | ---- | --- | --- | --- | --- |
> | Vanilla NMT | 0.656 | 0.675 | 0.503 | 0.623 | 0.695 | 0.709 | 0.584 | 0.533 |
> | Random Char Noise | 0.667 | 0.684 | 0.514 | 0.633 | 0.700 | 0.705 | 0.595 | 0.528 |
> | Unsupervised Noise Injection | 0.670 | 0.688 | 0.517 | 0.630 | 0.701 | 0.717 | 0.602 | 0.537 | KIWI
> | Supervised Noise Injection | 0.669 | 0.689 | 0.530 | 0.641 | 0.707 | 0.719 | 0.606 | 0.530 |
>
> COMETKIWI scores for Romance Family
> | Models | cat | glg | ast | oci |
> | -------- | ----- | ---- | ----- | ---- |
> | Vanilla NMT | 0.443 | 0.494 | 0.480 | 0.353 |
> | Random Char Noise | 0.495 | 0.568 | 0.562 | 0.403 |
> | Unsupervised Noise Injection | 0.499 | 0.584 | 0.581 | 0.407 |
> | Supervised Noise Injection | 0.495 | 0.586 | 0.584 | 0.403 |

---

### Meta-Review · Area_Chair_XSDg · 2023-09-12

**Recommendation:** 3

**Metareview:**

Reasons Accept:
The reviewers agree that this a simple and linguistically motivated data augmentation approach that could be straightforwardly applied to extremely low-resource language pairs.
Reasons Reject:
The proposed approach heavily relies on the existence of a high resource language which share linguistic features which limits its applicability. There could have been some analysis of how relatedness affects viability of the approach.
MetaReview
The authors submitted extensive rebuttal including new results which led to the increase in soundness from one reviewer from 3 to 4. All reviewers acknowledged rebuttal and there was some final discussion.

---

### Decision · Program_Chairs · 2023-10-07

**Decision:**

Accept-Findings

**Comment:**

Reasons Accept:
The reviewers agree that this a simple and linguistically motivated data augmentation approach that could be straightforwardly applied to extremely low-resource language pairs.
Reasons Reject:
The proposed approach heavily relies on the existence of a high resource language which share linguistic features which limits its applicability. There could have been some analysis of how relatedness affects viability of the approach.
MetaReview
The authors submitted extensive rebuttal including new results which led to the increase in soundness from one reviewer from 3 to 4. All reviewers acknowledged rebuttal and there was some final discussion.